# An adversarial collaboration protocol for testing contrasting predictions of global neuronal workspace and integrated information theory

**Lucia Melloni**[1,2‡]*, **Liad Mudrik**[3,4‡], **Michael Pitts**[5‡], **Katarina Bendtz**[6], **Oscar Ferrante**[7], **Urszula Gorska**[8], **Rony Hirschhorn**[4], **Aya Khalaf**[9,10], **Csaba Kozma**[8], **Alex Lepauvre**[1], **Ling Liu**[11,12], **David Mazumder**[6], **David Richter**[13], **Hao Zhou**[14,15], **Hal Blumenfeld**[8], **Melanie Boly**[16], **David J. Chalmers**[17], **Sasha Devore**[2], **Francis Fallon**[18], **Floris P. de Lange**[13], **Ole Jensen**[7], **Gabriel Kreiman**[6,19], **Huan Luo**[11,12], **Theofanis I. Panagiotaropoulos**[14], **Stanislas Dehaene**[14,20‡], **Christof Koch**[21‡], **Giulio Tononi**[8‡]

1 Neural Circuits, Consciousness and Cognition Research Group, Max Planck Institute for Empirical Aesthetics, Frankfurt am Main, Germany, 2 Department of Neurology, New York University Grossman School of Medicine, New York, New York, United States of America, 3 School of Psychological Sciences, Tel-Aviv University, Tel Aviv, Israel, 4 Sagol School of Neuroscience, Tel Aviv University, Tel Aviv, Israel, 5 Psychology Department, Reed College, Portland, Oregon, United States of America, 6 Children's Hospital, Harvard Medical School, Boston, Massachusetts, United States of America, 7 Centre for Human Brain Health, School of Psychology, University of Birmingham, Birmingham, United Kingdom, 8 Department of Psychiatry, University of Wisconsin-Madison, Madison, Wisconsin, United States of America, 9 Department of Neurology, Yale School of Medicine, New Haven, Connecticut, United States of America, 10 Biomedical Engineering and Systems, Faculty of Engineering, Cairo University, Giza, Egypt, 11 School of Psychological and Cognitive Science, Peking University, Peking, China, 12 IDG/McGovern Institute for Brain Science at Peking University, Peking, China, 13 Donders Institute for Brain, Cognition and Behaviour, Radboud University Nijmegen, Nijmegen, The Netherlands, 14 Cognitive Neuroimaging Unit, Commissariat à l'Energie Atomique (CEA), Institut National de la Santé et de la Recherche Médicale (INSERM) U992, Gif-sur-Yvette, France, 15 State Key Laboratory of Brain and Cognitive Science, Institute of Biophysics, Chinese Academy of Sciences, Beijing, China, 16 Department of Neurology, University of Wisconsin-Madison, Madison, Wisconsin, United States of America, 17 Department of Philosophy, New York University, New York, New York, United States of America, 18 Philosophy Department, St. John's University, New York, New York, United States of America, 19 Center for Brains, Minds and Machines, Boston, Massachusetts, United States of America, 20 Collège de France, Paris, France, 21 MindScope Program, Allen Institute, Seattle, Washington, United States of America

‡ LM, LM and MP share co-first authors on this work. SD, CK and GT are co-senior authors on this work.
* lucia.melloni@ae.mpg.de

**Data Availability Statement:** The data collected and analyzed for this Study Protocol will be made

## Abstract

The relationship between conscious experience and brain activity has intrigued scientists and philosophers for centuries. In the last decades, several theories have suggested different accounts for these relationships. These theories have developed in parallel, with little to no cross-talk among them. To advance research on consciousness, we established an adversarial collaboration between proponents of two of the major theories in the field, Global Neuronal Workspace and Integrated Information Theory. Together, we devised and preregistered two experiments that test contrasting predictions of these theories concerning the location and timing of correlates of visual consciousness, which have been endorsed by the theories' proponents. Predicted outcomes should either support, refute, or challenge these theories. Six theory-impartial laboratories will follow the study protocol specified here, using

openly available at OSF (https://osf.io/mbcfy/) and XNAT (Radiologics).

**Funding:** This project is made possible through the support of a grant from Templeton World Charity Foundation, Inc (TWCF0389) to Lucia Melloni and (TWCF0378) to Liad Mudrik and (TWCF0345) to Michael Pitts. The opinions expressed in this publication are those of the author(s) and do not necessarily reflect the views of Templeton World Charity Foundation, Inc. https://www.templetonworldcharity.org/. The funders had no role in study design, data collection and analysis, decision to publish, or preparation of the manuscript.

**Competing interests:** The authors have declared that no competing interests exist.

three complementary methods: Functional Magnetic Resonance Imaging (fMRI), Magneto-Electroencephalography (M-EEG), and intracranial electroencephalography (iEEG). The study protocol will include built-in replications, both between labs and within datasets. Through this ambitious undertaking, we hope to provide decisive evidence in favor or against the two theories and clarify the footprints of conscious visual perception in the human brain, while also providing an innovative model of large-scale, collaborative, and open science practice.

## Introduction

Understanding how consciousness relates to neural activity in the human brain remains one of the greatest scientific challenges [1]. Phenomenologically, consciousness has been defined as *subjective experience* (or *what it is like* to perceive, feel, act or think from a first-person perspective) [2]. In human experiments, consciousness is typically inferred based on the ability to report these experiences (either to oneself or to an external observer [3]). The neural correlates of consciousness (NCCs) have been defined as the *minimal* neuronal mechanisms jointly sufficient for any specific experience [4]. Significant progress has come from focusing on the search for the NCC [5, 6]. This has led to the development of several promising scientific theories of consciousness [7].

These theories, however, have evolved independently, without cross-talk. Empirical research has focused on testing theories separately, rather than on directly contrasting them to evaluate their explanatory and predictive power [8]. Dissatisfaction with this state of affairs has inspired an alternative model of research, in which adherents of competing views work together to derive experiments that directly compare predictions [9]. Dubbed *adversarial collaboration*, the goal is to reach an agreed-upon experimental design and to document protagonists' expectations concerning the outcomes of the experiment prior to the acquisition and analysis of the data [10, 11] (for a precursor to such an approach, see [12]). This approach can be combined with pre-registration of the research protocols and methods [13].

The present study protocol describes an adversarial collaboration involving two leading theories of consciousness: *Global Neuronal Workspace* (GNW) and *Integrated Information Theory* (IIT). The status of these theories as two of the most widely-discussed and well-supported in the field has been confirmed by recent reviews, meta-analyses, and surveys [14–16]. The project will focus on conscious vision in human subjects and involve two experiments, using three complementary methods in cognitive neuroscience: functional magnetic resonance imaging (fMRI), simultaneous magnetoencephalography & electroencephalography (M-EEG) and intracranial electroencephalography (iEEG). It will include built-in replications in two forms: first, for each method, data will be acquired from two independent laboratories. Second, the data will be split in half and the second half will be used for replication of analyses. In addition, the data and analysis protocols will be openly shared for reproducibility, and large sample sizes will be attained for statistical robustness. Taken together, these open science and collaborative practices should yield reliable results that can provide substantial evidence for one or the other theory, in order to arbitrate between them. For readers who wish to review the original preregistration and its amendments, detailing the experimental protocol and analysis plans, please see the OSF preregistration (https://osf.io/mbcfy/).

## Overview of the competing theories

*Global neuronal workspace theory* (GNW) posits that what we subjectively experience as a conscious state, at any given moment, is the global broadcasting and amplification of information [17] across an interconnected network of prefrontal-parietal areas and many distant high-level sensory cortical areas [18–20]. Unconscious processing occurs in parallel in many localized, modular circuits (e.g., in the ventral visual stream); if this processing ignites the global neuronal workspace (at about 250 ms post-stimulus presentation), information becomes conscious, being broadcasted and sustained by the workspace [18]. The workspace is constituted by a network of cortical neurons with long-range reciprocal projections to homologous neurons in other cortical areas, distributed over prefrontal (PFC), parieto-temporal and cingulate associative cortices. These neurons, mostly pyramidal cells of layers 2 and 3 (but also in layer 5), are connected through long-range excitatory axons to high-level sensory areas, allowing for flexible, domain-general amplification and distribution and exchange of information to various cognitive systems [21].

Anatomical tracer studies in non-human primates support the existence of a higher-order associative cortical network, with direct reciprocal connections between its component areas, that match the GNW core areas. Specifically, lateral prefrontal and posterior parietal cortex are connected through the superior longitudinal/arcuate fasciculus, while lateral prefrontal and superior lateral temporal cortex are connected through the extreme capsule fasciculus [22]. Parietal and temporal areas that are connected with prefrontal cortex are also heavily interconnected through the middle longitudinal fasciculus [23, 24]. Non-invasive diffusion tensor imaging studies indicate the conservation of these pathways in the human brain [22]. Importantly, these cortical areas that GNW proposes as critical for conscious information processing have been shown to comprise a highly interconnected cortical core, characterized by high information transfer efficiency [25]. This frontal-parietal-temporal network was found through graph theoretical measures to be at the top of a hierarchical cortical organization. Schematically, these areas can be viewed as being in the center of a bow-tie representation of interareal architecture and connected through feedback and feedforward loops with hierarchically lower areas in the periphery of this structure. GNW proposes that conscious access to a given information occurs when this central core network becomes invaded by neural activity coding for this information, which is achieved when this activity reaches a threshold for global ignition [26].

*Integrated information theory* (IIT) addresses consciousness starting from phenomenology—the existence of one's own experience, which is immediate and indubitable. As a first step, IIT identifies five properties of consciousness, which it calls "axioms", that are essential and true of every conceivable experience [27, 28]. The five axioms of IIT are intrinsic existence, composition, information, integration, and exclusion. Briefly, intrinsicality means that every experience is subjective—for the intrinsic perspective of the subject of experience, rather than for something extrinsic to it. Composition means that every experience is structured, being composed of phenomenal distinctions and relations, rather than without content. Information means that every experience is the specific way it is, rather than generic. Integration means that every experience is unitary, being irreducible to independent components, rather than multiple. Finally, exclusion means that every experience is definite—it contains what it contains, rather than having no definite border and grain.

IIT then seeks to provide an explanation for experience in physical terms, where "physical" has a precise operational definition in terms of cause-effect power—being able to take or make a difference. A physical substrate is simply a set of units that can be observed and manipulated, such as neurons that may be made to fire or not. In principle, the cause-effect

power of a physical substrate is characterized in terms of conditional probabilities: how the system responds to all possible perturbations of its state. On this basis, IIT's next step is to "translate" the essential phenomenal properties into essential physical properties, which it calls "postulates". In physical terms, intrinsicality means that the physical structure of consciousness (PSC) must have cause-effect for itself: it must be able to take and make a difference within itself. Composition means that its cause-effect power must be structured: it must have subsets of units that specify causal distinctions (causes and effects) bound by relations (overlaps among causes or effects), yielding a cause-effect structure. Information means that its cause-effect power must be specific, selecting for its subsets specific causes, effects, and relations. Integration means that its cause-effect power must be unitary: its cause-effect structure must be irreducible to that of its separate parts. Finally, exclusion means that its cause-effect power must be definite—its cause-effect structure must be specified by a definite set of units at a definite grain.

Much effort has gone into giving these postulates a mathematical form. In principle, this makes it possible to take a physical substrate, say a portion of our brain at a particular grain, and fully unfold its cause-effect structure. It follows from IIT that, to account for the essential properties of consciousness, the PSC must unfold into a specific cause-effect structure that is maximally irreducible. IIT introduces a scalar measure for integrated information ($\phi$ or phi), defined as the maximum of intrinsic, integrated cause-effect power over the substrate [29]. On this basis, IIT proposes its fundamental, explanatory identity: the components of the cause-effect structure specified by the PSC correspond one-to-one to the components of experience—to the quality of a specific experience. Thus, every content of an experience "here and now," including the experience of spatial extendedness, of time flowing, of objects and their local qualities, correspond to sub-structures in that cause-effect structure. Furthermore, the quantity of experience—how much one exists phenomenally—is measured by the irreducibility $\Phi$ of the cause-effect structure.

Thus, in IIT the NCC are the neuronal mechanisms at the relevant spatio-temporal level of granularity that maximize $\phi$ across the brain. Based on theoretical and neuroanatomical considerations, a substrate of maximum $\phi$ is hypothesized to reside primarily [although possibly not exclusively; 30] in the posterior cerebral cortex, characterized by 'pyramid-of-grids'-like connectivity. These regions, including the parietal, occipital and lateral temporal lobes, are referred to as the posterior "hot zone" [30].

## Differential predictions of the theories

This collaboration aims to test five distinctive predictions that can arbitrate between the two theories:

*1. Location of NCC*–GNW posits that every conscious experience is accompanied by activation of a fronto-parietal network in tandem with high-level sensory cortices. In contrast, an auxiliary prediction of IIT states that the NCC is primarily localized to the posterior hot zone. Though GNW refers to fronto-parietal networks, our study focuses on the NCC in the prefrontal cortex (PFC), as this is a main point of disagreement between the theories. Given contemporary experimental techniques, these differential predictions about NCC location remain the most viable and testable point of disagreement between the theories [30–32].

*2. Decoding content of consciousness*–GNW posits that information about the content of experience should be present both in the prefrontal-parietal network and high-level sensory cortices, as a key function of consciousness is global information sharing, implying consistent decodability of the content of experience in all of these regions. Conversely, for IIT, an experience is a structure, not a 'message' to be broadcast, with its content specified intrinsically by

the system and for the system, in the form of an integrated cause-effect structure made of distinctions and relations that are specified by the physical substrate of consciousness, presumed to be primarily located in posterior cortex. Thus, for IIT the contents of consciousness should be maximally decodable from posterior areas.

*3. Temporal dynamics of NCC*–GNW assumes that all experiences, regardless of their duration, are accompanied by an initial ignition (i.e., a non-linear activation marking the entrance of the information into the workspace). This activity, however, need not stay sustained and may be followed by a decay of signal back to baseline. This is because, in agreement with models of belief updating and predictive coding, the global ignition is thought to only reflect an update signal that refreshes the internal model. Such a refresh is needed only when new sensory information is not fully predicted by the internal model. During periods where sensory information is stable or can be fully predicted by the internal model (in the case, say, of a constant picture or a regularly repeated sound), then no refresh is needed, and the workspace is free to orient to other conscious thoughts. The maintenance of an internal model may occur in the form of an activity-silent state, with only occasional bursts of reactivation, similar to the concept of silent working memory [33–35].

Activity-silent working memory in the PFC has been described in both simulations and empirical data [e.g., 33, 36, 37]. It corresponds to the finding that, during the delay period of a working memory task, neural activity may not be continuously sustained, but return to baseline. Conscious working memory retrieval would occur only during short activity bursts, for instance towards the end of a working memory delay when content-specific spiking activity re-emerges ("ramps-up"). During this later period, the memorized item becomes task relevant, and the memory trace needs to be reactivated in order to elicit a response. During the activity-silent period itself, maintenance of information is thought to be supported by short-term synaptic weight changes through activity-dependent short-term synaptic plasticity. According to GNW, once an internal model has been refreshed by a transient ignition, the new model may be stored in a non-conscious silent state resembling activity-silent working memory.

According to predictive theories, any discrepancy between sensory inputs and their internal models causes the production of a transient prediction error signal [e.g., 38–40]. As a consequence, a brief ignition is also expected at the *offset* of the stimuli (if this offset is consciously detected) when there is also a prediction error that requires updating the conscious internal model. Thus, GNW does not assume that the neural workspace remains active throughout a durable conscious experience, but at moments when conscious refreshes occur.

For IIT, on the other hand, the substrate of an experience is the maximally irreducible cause-effect structure. IIT is an identity theory whereby the conscious experience is identical to the cause-effect structure, and the latter is determined by the physical substrate of consciousness in the brain [41]. Thus, in contrast to GNW, IIT predicts that the physical substrate of consciousness should persist over the duration of a conscious experience.

*4. Pre-stimulus activity*–the theories differ in explaining why a given stimulus fails to be perceived. For GNW, this happens either because stimulation is too weak to reach the global neuronal workspace, or because the workspace is in a refractory state as it is 'occupied' by other content. Thus, higher prestimulus activity in prefrontal and parietal areas should reduce the chances of a new stimulus being experienced. IIT, conversely, holds that it is the state of posterior cortical areas that determines the likelihood of a stimulus to be perceived. Higher pre-stimulus excitability within category-specific areas or greater synchrony between these regions and lower-level areas (e.g., V1/V2) should increase the chances of a new stimulus being experienced.

*5. Functional connectivity*—GNW postulates that global information sharing is mediated by a meta-stable state of long-range interareal interactions including prefrontal areas and category

specific areas, depending on the content of conscious perception. This pattern of information sharing can be assessed by measures of (gamma/beta) synchronization [42]. IIT conversely claims that conscious experience of a particular content depends on the pattern of activity in posterior cortex specifying a cause-effect structure composed of specific distinctions and relations. The existence of relations (bindings) among active units is reflected by enhanced interareal short-range synchrony. Here, each distinct conscious experience should lead to distinct patterns of synchronization between category-selective areas in posterior cortex and early sensory areas. Thus, while both theories posit an important role for interareal interactions, their predictions differ with respect to the areas for which synchronization should occur during conscious processing. Specifically, GNW predicts increased neural synchronization between nodes of the prefrontal cortex and category selective areas supporting the specific content of consciousness (e.g., FFA in the case of experiencing a face). IIT does not share the same commitments and posits increased neural synchronization between such category selective areas (e.g., FFA) and early visual areas (i.e., V1/V2).

## Overall rationale for testing the competing predictions

A challenge in studying the NCC is that parts of the neural activity differentiating seen from unseen stimuli relate to either precursors or consequences of consciousness, and therefore are not the NCC itself [43]. This is relevant since most studies depend on subjects' reports or judgements about the stimuli, leading to an overestimation of the true NCC [44, 45]. Since the predictions of both theories pertain to the true NCC, they should hold true regardless of task.

Our study includes manipulations to test if the predicted patterns are indeed task-independent. Two complementary experiments were designed to confirm some of the theories' predictions while challenging others. The first experiment will focus on the mechanisms underlying consciously perceiving an uncontroversially visible stimulus. It will examine predictions concerning brain areas and dynamics involved in conscious experience and the maintenance of that experience over time, for task-relevant and irrelevant content. The second experiment will use an attentional manipulation to render some stimuli invisible, despite having equal physical strength, enabling a direct comparison between neural processing of consciously seen and not consciously seen stimuli. The two experiments will complement one another as one focuses on the mechanism responsible for sustaining a stimulus in consciousness and disentangling the task-related effects from perception, while the other explores the mechanism underlying the differences between conscious and unconscious processing [46, 47]. Their combined results will determine the likelihood of the data under GNW or IIT predictions, as well as whether each theory passes, fails, or is challenged by the tests (see Fig 1).

To test the predictions, we will acquire fMRI and M-EEG data in neurotypical adults and invasive electrophysiological measures, iEEG, in patients with refractory epilepsy for both experiments. This combination of methods will provide the best set of tests for the hypotheses, considering acknowledged limitations of instruments available for human neuroscience. To minimize implicit bias, researchers without prior commitment to either theory will acquire and analyze the data in consultation with expert advisors. All data will be collected in duplicate, in two separate laboratories with two independent samples (a minimum of 50 subjects will be collected for each experiment for fMRI and M-EEG; 25 subjects will be collected for each experiment for iEEG for a total of 250 collected datasets in each experiment. Whenever feasible, subjects will participate in both experiments to allow for generalization across experiments (if not possible, additional subjects will be recruited).

**Analysis approach.** We have preregistered the experimental design, predictions and expected outcomes for each theory, while leaving open the specification of the analysis

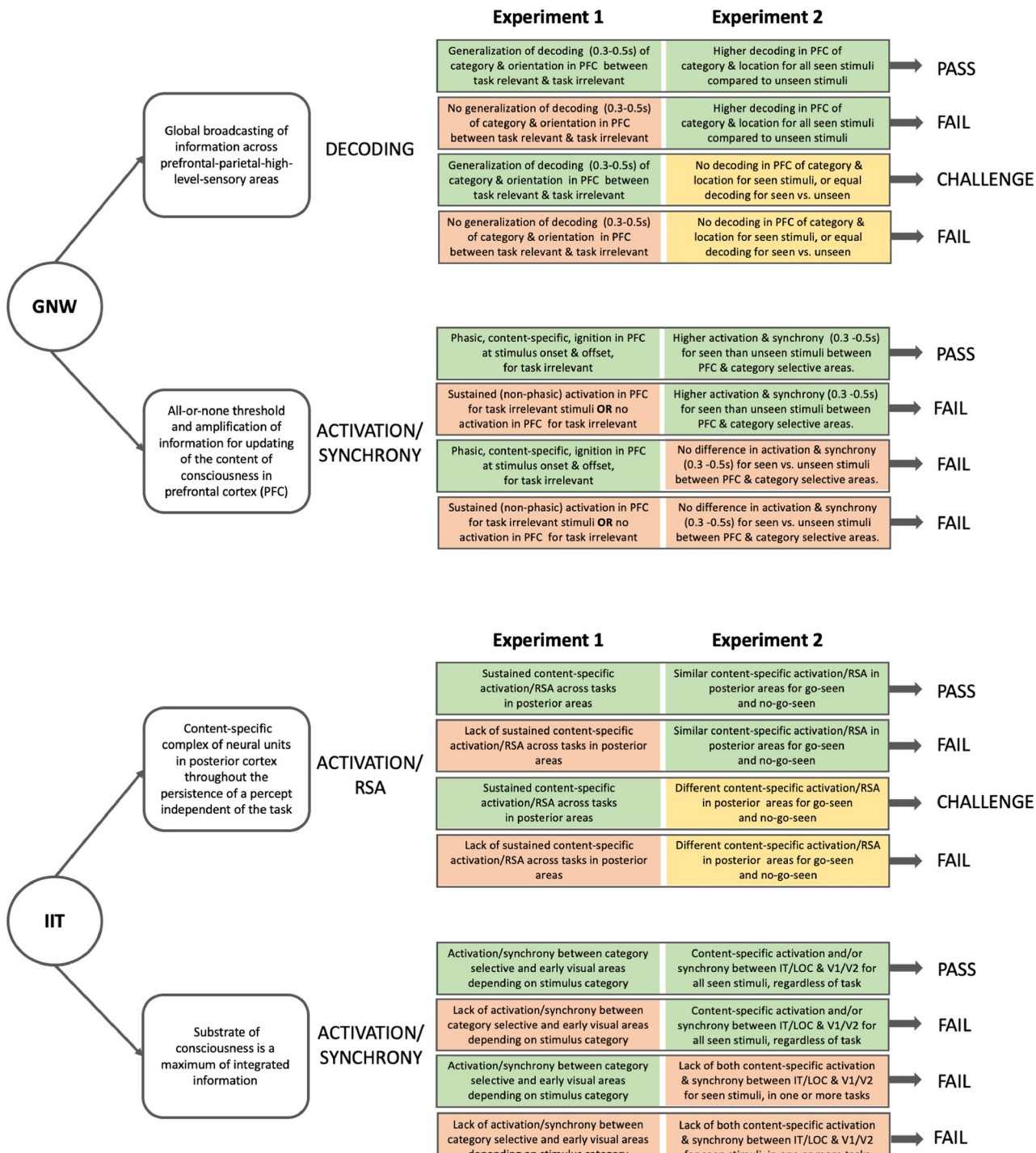

**Fig 1. Core predictions, analyses, and potential results for the two experiments.** For each theory (white circles), the core predictions (white rectangles) and possible outcomes (colored rectangles) are summarized. Green squares denote positive outcomes, red ones denote negative outcomes, and yellow squares denote negative outcomes that are less conclusive, as they might stem from limitations of the methods or analyses. On the right, we conclude how combinations of results would reflect on the theories. Note that some predictions require time-resolved measures (M-EEG, iEEG), while others can be tested with all three techniques. PFC refers to Prefrontal cortex; RSA refers to representational similarity analysis [RSA; 48]; IT refers to inferotemporal cortex; LOC refers to lateral occipital cortex; V1 refers to primary visual cortex.

methods. This will allow for the development of analyses while still committing to a preregistered protocol. The data will be split into two halves, so the first-half is used to optimize the analysis methods (phase 2, analysis optimization), with the aid of expert advisors. Then, analysis routines and all specific details will be locked, the preregistration will be amended, and the second half of the data will be used for replication (phase 3, replication). Thus, in accordance with our plan, the current protocol includes a higher-level description of the analyses and explicit, fleshed-out predictions, but does not include all of the analysis details.

**Statistical approach.** We plan to use a combination of classical and Bayesian statistics. The advantages of using Bayesian inference are threefold. First, it will enable us to assess the weight of evidence for each theories' prediction for the different analyses. Second, it will allow us to conclude that there is no evidence supporting a prediction (or for one theory over the other) when the data fails to show a predicted effect. This is a key point that contrasts with classical inference, where the null hypothesis of no difference can never be accepted. Third, it will allow us to assess the likelihood of the theories across the different analyses and different brain recording methods.

## Materials and methods

### Status and timeline

In early 2018, a group convened at the Allen Institute for Brain Sciences to map out contrasting predictions and ideas for experiments and methods. Over the course of the following months, the three first authors worked in conjunction with the leading theorists, and with input of the group, to specify experiments, instrumentation, recording methods, sample sizes, exclusion criteria, and analysis methods, which were all preregistered (https://osf.io/mbcfy/). During 2019, the three first authors carried out pilot behavioral and eye-tracking studies to refine and finalize the experimental designs. The initial preregistration was accordingly amended. While deploying and pilot testing the experiments in the six data acquisition labs, additional changes were introduced, and were again preregistered. The original and the amended preregistration files were frozen prior to the commencement of data acquisition.

The project is tripartite. At the time of submission of this study protocol, phase 1 is approximately halfway complete. In **phase 1**, all data will be acquired by theory neutral teams. To ensure replicability of the results, the entire dataset will be split into two halves, each with an equal mixture of data from each of the two labs for each recording technique (half of the participants tested in both labs). In **phase 2**, after evaluating data quality (see OSF preregistration), the first half of the data will be used for developing analysis tools (optimization of methods). The purpose of **phase 2** is to define the best analysis practices and to agree upon, in consultation with expert advisors, the detailed analysis protocols that will be preregistered in the final, unchangeable version of the preregistration. In our original protocol, phase 1 and phase 2 were planned to be conducted successively, yet due to delays in data acquisition caused by the COVID-19 pandemic of 2020–2021, now phase 1 and phase 2 will overlap. In **phase 3**, the replication phase, the second half of the data will be analyzed using these preregistered, agreed upon protocols, thereby allowing an in-house replication of the results obtained in phase 2. Data will be analyzed by the acquiring laboratories, a data analysis team composed of the two theory leaders, the center PIs (the three co-first authors leading this project), and a method-experts advisory panel. All data, meta-data, stimulation protocols, and analysis pipelines will be publicly released upon the publication of the final results via a combination of documents in OSF, raw data, processed data and analysis code, and protocol instructions and specifications in SLAB (data management and sharing plan are detailed below).

## Experiment 1

**Aim.** Experiment 1 will test several predictions of GNW and IIT (detailed below): the most critical for GNW relates to its predictions concerning information about a clearly visible stimulus, the decodability of which should not depend on whether the percept is task-relevant or not; and IIT's critical prediction that the physical substrate of consciousness should remain active throughout the presentation of a clearly visible stimulus, whether task-relevant or not.

**Design, stimuli, and procedure.** We will measure brain activity elicited by visual stimuli that are clearly consciously perceived while manipulating (1) the task, such that some stimuli are task-relevant while others are task-irrelevant [following 49], (2), stimulus duration, such that stimuli are perceived for different durations [50], (3) stimulus category, such that the content of perception varies, and (4) stimulus orientation, such that the specific features of the perceived content within a given category can be tested under fully task-irrelevant conditions.

We will use four categories of stimuli that naturally fall into two distinct classes—pictures (20 faces, half male, half female, and 20 objects) and symbols (20 letters and 20 false-fonts). Stimulus orientation will be manipulated, *i.e.*, half will have a side view (rotated +/- 30˚) and half a front view. All stimuli will be supra-threshold, presented at fixation and subtend the same size in visual angle (6˚x6˚).

In each 2s trial, one stimulus from one of the four categories will be presented, in greyscale, for a duration of 500, 1000 or 1500ms, followed by a blank. The experiment will be divided into mini-blocks, so that for each mini-block, half the trials will contain task-relevant stimuli and the other half task-irrelevant stimuli. To define task-relevance, subjects will be instructed to detect the occurrences of two targets regardless of their orientations (targets in both orientations will be presented at the beginning of the trial) belonging to two different categories (counterbalanced between mini-blocks) (Fig 2). Accordingly, each mini-block will contain three different trial types: i) *Task-Relevant Targets*: the two stimuli being detected in that block (e.g., a specific face and object; upper row in Fig 2; or a specific letter and a false-font; lower row in Fig 2); ii) *Task-Relevant Non-Targets*: stimuli from the task-relevant categories but not the specific targets (e.g., other faces and objects; highlighted in blue in upper row in Fig 2); and iii) *Task-Irrelevant Stimuli*: stimuli from the other categories (e.g., letters and false-fonts; highlighted in green in upper row in Fig 2). Thus, the very same stimuli will be task-relevant non-targets in some blocks and task-irrelevant in other blocks. Subjects will be asked to maintain central fixation throughout each trial. Gaze will be monitored online through an eye tracker.

Thus, in formal terms, Experiment 1 is a nested factorial design with four factors: stimulus category (face, object, letter, false-font), task relevance (task-relevant targets, task-relevant non-targets, task-irrelevant stimuli), stimulus duration (500, 1000, 1500 ms), and stimulus orientation (front view, side view). We are particularly interested in the interactions between stimulus category and task-relevance, as well as between stimulus duration and task-relevance. Trial numbers, stimulus randomization and timing parameters are optimized for the different methodologies i.e., fMRI, M-EEG and iEEG. Technique specific details are described in the preregistration in OSF.

**Participants, sample size, exclusion criteria and stopping rule.** For the fMRI and M-EEG studies, all participants will be older than 18 years old, have reportedly normal or corrected-to-normal visual acuity and no known history of psychiatric or neurological disorders. For the iEEG studies, subjects will be 10–65 years old, be able to provide informed consent, have IQ > 70, be fluent in English or Spanish, self-reported normal hearing, normal or corrected-to-normal vision, and show cognitive and language abilities within or above the normal range in formal neuropsychological testing performed before surgery and must not have had

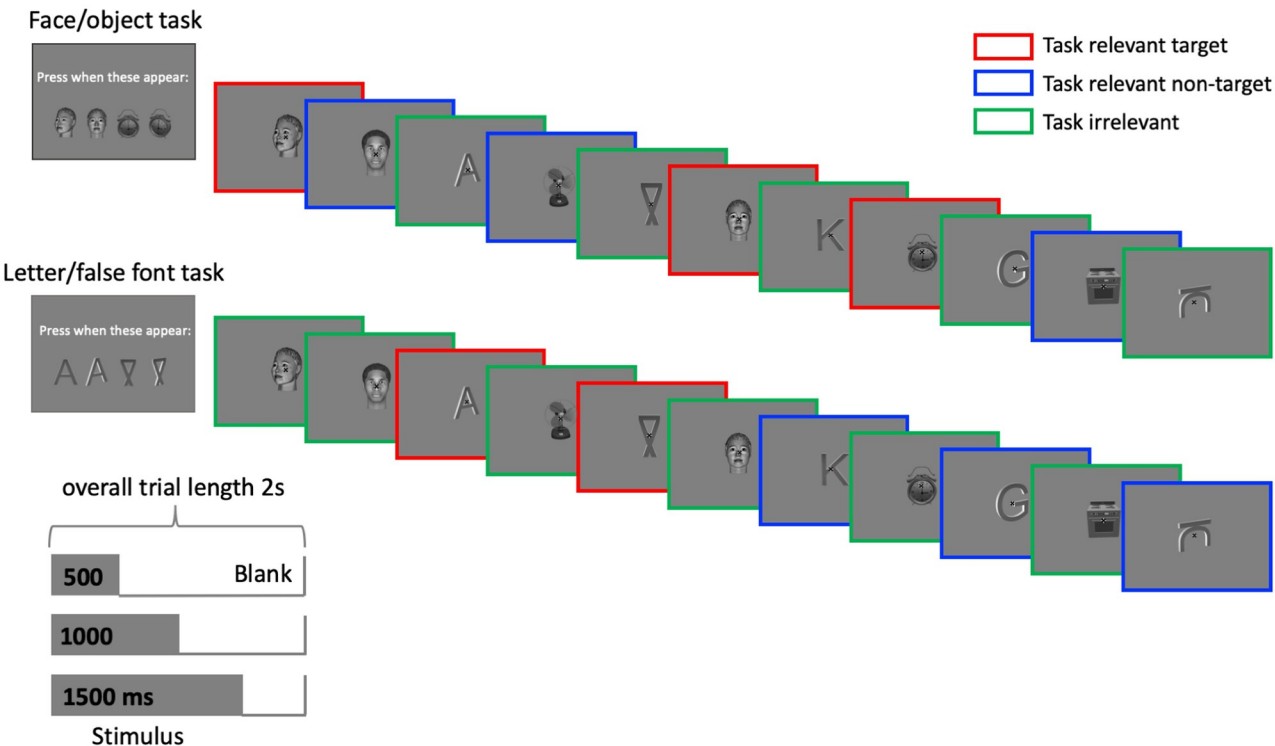

**Fig 2.** Experiment 1 design—In separate blocks (top & bottom rows), subjects will be asked to detect infrequent targets drawn from two categories: a specific face and a specific object (upper row), or a specific letter and a specific false-font (lower row). The stimuli will be shared across blocks, but the selection of the targets will determine to which of three trial types each stimulus belongs: *task-relevant targets*, *task-relevant non-targets*, *task-irrelevant non-targets* highlighted by red, blue and green frames respectively. Colored frames are shown for illustration purposes only. The duration (500, 1000, 1500 ms) and orientation (side view/front view) of stimuli will be manipulated to allow for specific analyses. Blank intervals between stimuli and intertrial intervals (truncated exponential distribution) are not depicted here.

an electrographic seizure within 3-hours prior to testing. When available, information concerning language dominance as assessed by the intracarotid sodium amobarbital (Wada) will be reported.

Informed consent will be obtained prior to the study. In the case of minors, assent will be obtained from the minor and informed consent will be obtained from a parent or a legal guardian. IRB approval has been obtained from each local site in which the experiments are being conducted. Patients will be informed that participation in the study will not affect their clinical care and that they could withdraw their participation from the study at any point without affecting their clinical care. All subjects will be informed that data will be made publicly available for a minimum of 5 years after the publication of the main results. Only subjects agreeing to the data sharing procedure will be recruited.

Targeted sample sizes were determined for each methodology as being 2.5 times larger than common sample sizes in the literature for that methodology (50 subjects for fMRI and for M-EEG given that 20 subjects is the common sample size; 25 subjects for iEEG given that 10 subjects is the common sample size). Samples sizes refer to total data that will be collected in each individual laboratory, i.e., total sample sizes when the data is combined across labs will be one hundred for fMRI, one hundred for M-EEG, and fifty for iEEG. We estimate a 20% attrition related to data quality. Even after considering attrition, since we will use a within-subject design, this sample size will give us >90% power to detect differences of medium effect size (Cohen's d = >0.5).

The data will be analyzed by the data monitoring team (who are not part of the data acquisition labs) to ensure data quality. This will be done both behaviorally, aimed at excluding subjects who do not meet our predefined criteria (see below), and physiologically, to test data quality (e.g., by assessing signal to noise ratio in the neural data, eye movement patterns in the eye tracking data, etc.). These data quality checks are orthogonal to the tested predictions by the theories. We will stop collecting data upon completing the predefined sample size. No further rejection of subjects will occur after this stage.

The exclusion criteria will include: (a) insufficient number of trials in each of the experimental conditions, due to excessive eye movements, muscular artifacts, movement, noisy recording, or subjects deciding to stop the experiments (<30 for M-EEG or <20 for fMRI. If analyses show that a good enough signal can be obtained with fewer trials, these numbers will be amended); and (b) low performance in the attention tasks. In Experiment 1, this translates into: <80% hits, >20% FAs for fMRI and M-EEG subjects; <70% hits, >30% FAs for iEEG patients. For Experiment 2 performance criteria, see below.

## Overview of operational hypotheses of the theories

Based on the opposing predictions of the theories, we will focus on testing three predictions. The first refers to **decodability of content of consciousness** (see analysis #1 below). Decoding will be tested for category as well as for orientation (which is a task irrelevant feature, because–as opposed to category–it is completely orthogonal to the task). As the stimuli are clearly visible in this experiment, the theories predict that information about the content of consciousness should be consistently found in the underlying neural substrates, even if the stimulus (or one of its features, like orientation) is task-irrelevant. Thus, according to GNW, such information should be found in nodes of prefrontal areas, while according to IIT it should be primarily found in posterior areas. The critical test of these predictions will accordingly be whether decoding of category and orientation generalizes from the task relevant (non-target) condition to the task irrelevant condition, within the predicted brain regions (PFC for GNW versus posterior hot zone for IIT). The second prediction refers to the **maintenance of a percept over time**, manipulated here via stimuli duration (see analysis #2 below). GNW predicts transient activity, reflecting an updating mechanism at stimulus onset and offset, while IIT predicts that the physical substrate of consciousness should remain in a similar activity pattern throughout the duration of the conscious experience (here, as long as the stimulus was presented). The third prediction focuses on **inter-areal communication** between different cortical regions during conscious perception (see analysis #3 below). GNW and IIT postulate different nodes (topologies) related to consciously perceiving a stimulus: for GNW, long-range synchrony is predicted between prefrontal and category-selective areas in high-level sensory cortex; for IIT synchronization is predicted within the posterior hot-zone, between category-selective regions in high-level sensory cortex and lower-level visual areas. Finally, an additional goal of this experiment is to **pinpoint the putative NCC**, while detecting areas that cannot be regarded as such, as their activity reflects task related processes (see analysis #4 below). This holds implications for the theories, as they differ in their predictions about the NCC (see'*Neural Activations related to consciousness*' above).

## Planned analyses and expected outcomes

**1. Multivariate decoding.**   To test which cortical areas contain information about clearly perceived stimuli across both task-relevant and task-irrelevant conditions, we will decode stimulus category (face/object or letter/false-font) and orientation (side view/front view) in the task-relevant non-target condition and then test for cross-task generalization of decoding to

the task-irrelevant condition. Cross-task generalization from task-irrelevant to task-relevant condition will also be tested. Category decoding will always be carried-out within each stimulus class (faces vs. objects and letters vs. false-fonts), rather than between classes which would confound category and task (i.e., we will not decode faces vs. letters, or objects vs. false-fonts). Note that as opposed to information on category, which will sometimes be relevant and sometimes irrelevant, information on orientation will always be task-irrelevant during the entire experiment, despite being clearly seen. As mentioned above, here and in all analyses, we will use the first half of the data to optimize our analyses during phase 2 of the project (analysis development stage), including the selection of optimal decoding schemes (options include: linear SVMs, Bayesian SVMs, LDA, naïve Bayes, K-nearest neighbor, logistic regression, random forest).

GNW predicts that conscious content will be decodable from areas in the prefrontal cortex i.e., inferior frontal gyrus, middle frontal gyrus, and cingulate cortex [21, 51, 52], instantiating the global workspace, both for task relevant and task irrelevant conditions. Thus, GNW would expect transient information about category and orientation to be decodable from the PFC and global-workspace recipient regions in posterior and higher-order sensory cortex in task-relevant conditions, and for decoding to generalize between the task-relevant and task-irrelevant conditions and vice versa, specifically during the 300–500 ms time window. IIT predicts that content-specific NCCs will primarily be found in the posterior hot zone, independent of task-relevance, and makes no specific predictions as to the relevant time window. Thus, decoding of categories and orientations and cross-task generalization of decoding between the task-relevant and task-irrelevant conditions and vice versa should be maximal in posterior cortical areas.

**2. Activation patterns across stimulus durations.** This analysis will target the relation between the temporal duration of the experience and the neural responses. Since subjects will observe clearly visible isolated stimuli at fixation and without competition, we assume that they will experience the stimuli for as long as they are on-screen, in particular because subjects' fixation (via eye-tracking) will be constantly monitored (see the OSF page for a control experiment evaluating visibility of long-duration task irrelevant stimuli).

Here, we will evaluate the neural responses associated with the maintenance of a percept in consciousness. Specifically, we will investigate the temporal profile of activation evoked by stimuli of different durations as well as the temporal profile of informational content of the neural responses throughout the duration of the stimulus. For the former, we will compare activation levels between conditions (defined by stimulus duration) in four different time windows: (1) 300–500 ms, (2) 800–100 ms, (3) 1300–1500 ms, and (4) 1800–2000 ms using linear mixed models. For the latter, we will combine temporal generalization [53] and representational similarity analysis [RSA; 48] approaches: in long duration trials, stimulus category (as well as orientation) will be decoded in a temporal generalization fashion (testing for generalization of neural patterns containing the content specific information across time). Representational similarity analysis will then be employed to compare the resulting temporal generalization matrix to the predictions of the theories in the specified time windows.

GNW predicts ignitions within PFC following stimulus onset and possibly also during stimulus offset, constituting updates to conscious perception, with virtually no activity between updates [except for occasional stochastic bursts of spontaneous reactivation; 54]. Thus, for consciously perceived but task-irrelevant stimuli, neural activation is predicted to reflect an interaction between duration and time window in the PFC, such that activation is maximal in windows following stimulus onset and offset and minimal elsewhere. GNW further predicts that the patterns of activation will be consistent and content-specific between the windows where activation is expected to be maximum, whereas in the windows of minimal activation,

there should be no content-specific activation. This translates in expected temporal generalization of decoding accuracy between the onset and offset burst with no generalization in between. This prediction can be translated in a model matrix to be correlated with the obtained temporal generalization results. IIT predicts sustained, content-specific activity patterns in posterior cortex throughout the duration of the percept (after the initial transient response). Therefore, activation levels within posterior areas should be higher for longer-duration (and consciously seen) than shorter-duration (and no longer seen) stimuli during analysis time windows 2 and 3. In addition, temporal generalization of decoding accuracy should be evident for as long as the stimuli are consciously experienced, and will be tested via correlation with a different model matrix than for GNW.

**3. Inter-areal communication: Patterns of synchronization.**   This analysis will focus on information sharing/integration between different cortical regions for each specific stimulus category.

GNW predicts that conscious perception, regardless of task, relies on long-range information sharing between nodes of PFC and Fusiform Face Area (FFA) when seeing a face, and PFC and Lateral Occipital Cortex (LOC) when seeing an object. This implies that synchronization within these regions should be more consistent across tasks (for a given stimulus category) than across stimuli (for a given task). IIT predicts strong bindings ('relations') between category-selective areas and lower-level visual areas, which again should be found regardless of the task. Hence, patterns of synchronization should be found between category-selective regions (FFA and LOC) and early visual cortex: specifically, between FFA and V1/V2 when seen faces (but not other stimulus categories) and between LOC and V1/V2 when seen objects (but not other stimulus categories). This synchronization should be more consistent across tasks than across stimuli.

**4. Putative NCC analysis.**   We will also perform further analyses to delineate the putative NCC for which GNW and IIT provide different predictions (PFC vs. posterior hot zone, respectively), after ruling out areas based on task contrasts (see the OSF preregistration for more details). To that end, we will run three contrast/conjunction analyses, which can be performed on univariate fMRI activation maps as well as on multivariate fMRI decoding maps: (A) Areas that are sensitive to *task goal (attending, detecting, & responding to target stimuli)*, showing greater activity for task-relevant targets vs. baseline (blank inter-trial intervals), and–importantly–no differential activity for non-targets (both for task-relevant or task-irrelevant) vs. baseline (blank ITIs); (B) Areas that are sensitive to *task-relevance (attending & detecting stimuli of the relevant categories)*, and accordingly are responsive to all task-relevant stimuli, whether targets or not, but are not responsive to task-irrelevant stimuli; (C) *candidate areas for enabling conscious perception (putative NCCs)*. We will look for the conjunction of areas sensitive to changes in the content of consciousness (stimulus present vs. blank ITIs) within each stimulus category (e.g., faces vs. blank, objects vs. blank, letters vs. blank, false fonts vs. blank).

The first two analyses (A & B) are aimed at *identifying areas that are most likely involved in the consequences of consciousness and are unlikely to be related to neural processes mediating consciousness per se*. The third analysis (C) is aimed at identifying *areas that may contain NCCs and/or sensory precursors to visual consciousness*. We note that the proposed contrasts might underestimate the consequences of consciousness (analysis A and B) and overestimate the NCC (analysis C). We have adopted a conservative approach which has the advantage of firmly, while not exhaustively, distinguishing between areas that might participate in consciousness vs. those that definitely do not. GNW predicts that the putative NCC should include prefrontal cortex after ruling out areas based on task contrasts. IIT in turn predicts that the putative NCC should be mapped primarily onto posterior areas, after ruling out areas based on task contrasts.

*Pilot data*. To establish the adequacy of our experimental protocols, this experiment has been pilot-tested for behavioral and eye movement patterns. The results of these pilot studies are reported on the OSF page (https://osf.io/6vgy3/). We also conducted a control experiment testing the visibility of the stimuli in an offline, surprise memory test. These pilot tests showed that subjects detect the targets with high sensitivity and specificity while being able to maintain fixation on the stimulus throughout the duration of the trial. They further confirmed that subjects were indeed aware of all stimuli, including the task-irrelevant ones, as performance and confidence for task-irrelevant stimuli was overall similar to task-relevant stimuli.

## Experiment 2

**Aim.** The general goal of Experiment 2 is to measure brain activity elicited by salient visual stimuli that are reported as seen versus unseen in the context of a secondary task. Stimulus visibility is manipulated via attention, by using an engaging video game as a primary, attentionally-taxing task. As such, this experiment predominantly evaluates the neural mechanisms underlying conscious vs. unconscious processing. Several predictions of GNW and IIT will be tested. GNW's hypotheses are directly related to the contrast between reported-seen and reported-unseen trials, while IIT's predictions focus on seen trials across different task manipulations. The most critical tests of GNW relate to the decodability of conscious content in prefrontal areas for seen stimuli, regardless of task manipulations, as well as the prediction that ignition of the global workspace (information sharing)–as measured by long-range synchrony between the prefrontal and sensory cortices–is key to conscious perception (again, regardless of the task). IIT's central prediction in this experiment is that activity and patterns of connectivity within posterior cortical areas will be consistent across all task conditions for seen stimuli triggering similar experiences (within the same category), but different for seen stimuli triggering different experiences (across different categories), independently of the task in which those seen stimuli are embedded.

**Design, stimuli, and procedure.** We seek to measure brain activity elicited by salient stimuli reported as seen versus unseen due to inattentional blindness [55], in an attentional-taxing, engaging video game context. To minimize the relevance of the critical stimuli, subjects will be challenged to maximize performance on the video game, with visibility reports serving as a de-emphasized secondary task. This experiment will focus on investigating differences in activation, decoding, and level of inter-regional synchronization between seen and unseen stimuli, including prestimulus responses as well as early vs. late post-stimulus responses.

Large (2.3°), high contrast, faces and objects will be presented for 250 ms at one of four locations (two in the left, two in the right visual field, 6.4° eccentricity), while scrambled texture patterns [56] made from superimposing faces and objects will be presented at the other three locations. The stimuli will be displayed on top of rotating square shapes in the background of a video game, which spans the entire screen (to estimate the relative size of the stimuli compared with the background, see Fig 3).

Identical stimuli will be presented during a *distracted attention* (dAT) and an *attended* (AT) task. In the former, subjects will be instructed to exclusively focus on playing an engaging video game (for a detailed description, see the preregistration and the demo movie: https://osf.io/b9nce/) while infrequent "probes" will assess whether they consciously perceived a subset of the face and object stimuli presented in the background (see Fig 3 for further description of the videogame). The probes will be presented while the game is paused with an arrow cue pointing towards one of the potential stimulus locations. Subjects will be instructed to respond yes/no to each probe by pressing one of two keys to indicate if they saw a stimulus at that location or not. There will be two types of probes. "Awareness probes" will follow a stimulus presentation and

**Fig 3.** Experiment 2 design–A) The four worlds of the game, with four levels in each world (left). In worlds 1 & 3 (middle) the player will control a shiny blue orb at the bottom of the screen to collect falling disks of the same color (on three vertical tracks), while avoiding falling disks of a different color. The size of the full game display in visual angle is indicated here (panel A, middle). In worlds 2 & 4 the colors will be swapped and players will control an orange orb (right). B) Examples of stimuli presented in the background (central portion of screen blown-up for display purposes): face (left), object (middle), blank (right). C) During gameplay, at 3-6sec intervals (4-7sec for fMRI), an object/face/blank will appear on one of four background squares (middle) for 500ms total, 250ms at full contrast, followed by 250ms of fade-out. On some trials (every 9-18sec), the stimulus will be immediately followed by an awareness probe (right) in which the game will pause and a small arrow will appear in the center instructing subjects to report whether they had just noticed a stimulus in that location. D) Schematic of the background animation demonstrating the trial timing, stimulus timing, and probe timing. For a video demonstration of the video game, visit the preregistration website: https://osf.io/b9nce/).

**Table 1. Trial types.**

| Trial type | Label | Trial description |
|---|---|---|
| DISTRACTED-ATTENTION Reported-Seen | dAT-seen | Faces/objects reported as "seen" in the awareness probes |
| DISTRACTED-ATTENTION Reported-Unseen | dAT-unseen | Faces/objects reported as "unseen" in the awareness probes |
| DISTRACTED-ATTENTION Blanks | dAT-blank | Background changes identical to face/object trials, but with no face or object presented, and correctly reported as "no" in the awareness probes |
| ATTENDED Seen-Go | AT-seen-go | Faces/objects in the AT task that are the target category eliciting a go response |
| ATTENDED Seen-Nogo | AT-seen-nogo | Faces/objects in the AT task that are not the target category and thus elicit a no-go response |
| ATTENDED Blanks | AT-blank | Background changes identical to face/object trials, but with no face or object presented in the AT task |

the arrow will always point to the location of the most recently presented face or object. "Catch probes", on the other hand, will follow trials in which no stimulus was presented and the arrow will point to a random location (avoiding a location where a scrambled texture appeared, i.e., a "blank" location). The purpose of the retrospective location cues [57–60] is to minimize potential forgetting of the stimuli that happened to be consciously seen. Game difficulty will be adapted online based on performance, to ensure subjects' continuous engagement.

In the *attended* (AT) condition, the same visual display as in the dAT condition will be presented, serving as a 'replay' of the video game, but with the opposite instruction: ignore the video game, and "be on the lookout" for either faces or objects appearing in the background. Eight different runs of the AT condition will be performed: during four of these runs, subjects will be instructed to press a button when they see a face, but not when they see an object, such that faces elicit a "go" response and objects a "no-go" response (and vice versa). We assume that in the AT condition stimuli will be easily seen, as they are presented for a long duration, and subjects' only task will be to detect these stimuli (rather than playing the game as in the dAT condition). The crossing of the go/no-go task in the AT condition will yield two different trial types: AT-go-seen, and AT-nogo-seen, defined for both categories of stimuli (faces and objects). Both dAT and AT tasks will consist of several short runs, to allow for training and testing of classifiers on independent runs. The AT condition will thus serve both to define the ROIs for the dAT condition, and as an independent condition, to enable tests of generalization of decoding across the AT and dAT conditions.

Formally, Experiment 2 uses a nested factorial design with three factors: stimulus category (faces, objects), stimulus location (left, right), and task (distracted attention, attended). An additional factor will be measured but not manipulated: visibility (seen, unseen), which will be based on subjects' reports in the distracted attention condition. We are particularly interested in the interactions between stimulus category (and location) and visibility, as well as stimulus category (and location) and task. A summary of trial types is presented in Table 1. Trial numbers, stimulus randomization and timing parameters will be optimized for the different methodologies i.e., fMRI, M-EEG and iEEG. Technique specific details are described in the preregistration in OSF.

**Participants, sample size, exclusion criteria and stopping rule.** Whenever possible, we aim at using the same subjects and sample sizes in both experiments. As in Experiment 1, the data will be monitored for quality during the data collection phase of the project, and we will stop collecting data upon completing the pre-defined sample size.

We will use comparable exclusion criteria as in Experiment 1. Here subjects will be included if performance in the AT task is: >80% hits, <20% FAs for fMRI and M-EEG

subjects; >65% hits, <40% FAs for iEEG patients or if there is at least a 25% increase in hit rates in the AT task vs. seen rates in the dAT task. In addition, subjects will be excluded if they have too few seen/unseen trials in the dAT condition to conduct proper analyses (for a given stimulus category, if seen or unseen trial counts are <30 for M-EEG or <20 for fMRI. If analyses show that a good enough signal can be obtained with fewer trials, these numbers will be amended).

**Behavioral pre-screening.** Based on extensive behavioral pilot testing, inter-subject variability in the rates of stimuli reported as seen/unseen is expected. To ensure a sufficient number of trials for the planned analyses, subjects in the fMRI and M-EEG studies will participate in a behavioral pre-screening session (prior to recruitment for Experiment 1). In this pre-screening, subjects will play the first world of the game, which includes 50 awareness probes (40 faces/objects and 10 blank catch- trials). Subjects who report seeing between 5–32 faces/objects (12.5–80% of probed stimuli) and report seeing stimuli on 4 or fewer blank catch-trials (40% or less) will pass the pre-screening and will be recruited for participation in the full experiments. This procedure is designed to screen- out subjects who consciously perceive too many or too few of the critical stimuli (thus preventing a full analysis of the brain data due to insufficient numbers of trials) as well as subjects who are not careful enough about reporting what they see (i.e., subjects with too many false alarms). Considering the rareness and opportunistic sampling of the population of epilepsy patients undergoing invasive monitoring, no pre-screening will be applied in that sample. Behavior in patients is however likely to differ from neurotypicals. Given performance in collected datasets, if necessary, the experiment will be optimized, or else discontinued if behavior does not meet the expected range.

## Overview of operational hypotheses of the theories

Four different hypotheses will be tested in this experiment. The first set of predictions relates to **levels of activations and inter-areal communication** for stimuli reported as seen vs. unseen in the dAT task (see analysis #1 below). GNW proposes that a network of prefrontal and high-level sensory areas should be found for this contrast. Since opposing predictions between GNW and IIT concern mostly prefrontal cortex, we focus the analyses and predictions on this area. Note however that as per GNW prefrontal and parietal cortices as well as the cingulate are part of the global neural workspace. IIT postulates that differences should be present in posterior cortex between category-selective areas and early sensory areas (per IIT, prefrontal cortex activity is not necessary for consciousness, but differences in prefrontal activity may be found in the dAT condition, given that this is a report paradigm with a secondary target detection task). As for inter-areal communication and synchrony, we aim to test the role of integration, and more specifically the areas over which information is integrated/distributed for conscious percepts. Like in Experiment 1, GNW and IIT postulate different nodes (topologies) related to consciously perceiving a stimulus: when subjects report perceiving faces, GNW predicts long-range synchrony between PFC and Fusiform Face Area (FFA); and for consciously perceiving objects, GNW predicts long-range synchrony between PFC and Lateral Occipital Cortex (LOC). The specific topological pattern of synchronization should be present regardless of the task (although its timecourse may be prolonged under task-relevant conditions). Thus, the content specific pattern of synchronization is expected to be found both for stimuli reported as seen in the dAT and for seen stimuli in the go/no-go conditions in the AT tasks. It should, however, be absent on "unseen" trials of the dAT condition, except possibly for a brief and early (earlier than 300 ms) transient (failed ignition). For IIT, on the other hand, if a stimulus such as a face is seen, face selective cells in FFA should be activated and there should be strong bindings ('relations' in IIT) among neural units in posterior cortical areas, from FFA

down to earlier visual areas (V1/V2), which should translate into synchronization between these areas. By contrast, when an object is seen, object-selective cells in LOC should be activated and there should be strong bindings from LOC down to earlier visual areas (V1/V2), which can be measured as increased synchronization between these areas. These patterns should be present when stimuli are reported as seen in the dAT condition as well as in the go/no-go conditions of the AT task.

A second set of predictions tests **in which brain areas the content of conscious perception can be decoded** (see analysis #2 below). We will first test decoding of faces vs. objects in the AT task, and then subsequently test decoding of faces vs. objects in the dAT-seen and dAT-unseen conditions separately. Location of the stimulus (left/right visual field) will also be decoded first in the AT task and then for dAT-seen and dAT-unseen conditions separately. GNW predicts that conscious content (i.e., category and location) should be decodable from PFC during the time period of ignition (~300–500 ms), while unconscious content should not be decodable from these areas during this time period. IIT predicts that conscious contents should be best decoded from posterior cortex and with activity patterns that should be similar for seen stimuli across the go and no-go conditions in the AT task and for stimuli reported as seen in the dAT task.

A third prediction focuses on the **temporal dynamics of the process leading to a conscious percept** (see analysis #3 below). According to GNW, during the first 250 ms, activity should be similar in reported-seen and reported-unseen trials. After 250 ms, an ignition marks the activation of the global workspace by the stimulus, so higher activation in prefrontal areas for reported-seen trials, compared to reported-unseen ones, should be found during this later time-window. This prediction will only be tested for GNW, as it does not have direct bearings on IIT. Relatedly, we will test a fourth set of predictions about **the likelihood of a stimulus to be consciously perceived** (see analysis #4 below). GNW predicts that higher pre-stimulus activity in PFC should be found in reported-unseen trials compared to reported-seen trials. IIT, conversely, holds that higher pre-stimulus excitability or greater synchrony within the posterior hot zone should increase the chances of a new stimulus to be perceived.

## Planned analyses and expected outcomes

**1. Levels of activation and inter-areal communication.** GNW postulates that seen relative to unseen stimuli should activate a network of prefrontal and high-level sensory areas, while IIT avers that differences between seen and unseen stimuli should be present in the posterior hot zone, involving category-selective (FFA, LOC) and early sensory areas. We will test those anatomical predictions by evaluating the differential activity in these areas and the synchrony between them. This will be done by comparing the dAT-seen and dAT-unseen trials with an ROI and whole brain approach, separately for faces/objects and blank trials. We will calculate the synchrony between prefrontal ROIs and posterior category-selective areas, as well as between such category-selective regions and early visual areas (V1/V2), for: (a) go/no-go conditions separately, per stimulus category, vs. blanks in the AT task; and (b) dAT-seen and dAT-unseen separately, per stimulus category, in the dAT condition.

GNW and IIT postulate different nodes related to consciously perceiving a stimulus: when subjects report seeing faces or objects, GNW predicts long-range synchrony between PFC and FFA or LOC, respectively. For IIT, activation and mid-range synchrony between FFA or LOC and early visual areas (V1/V2) is expected for seen, but not for unseen trials. For both theories, the patterns of activation and synchronization should be present both when stimuli are reported as seen in the dAT condition and in go/no-go conditions of the AT task, but absent when the stimuli are not seen.

**2. Multivariate decoding of category and location—Activation patterns.** This analysis will test decoding of faces vs. objects, and left vs. right locations, in the AT, dAT-seen and dAT-unseen conditions, both with and without the ROI for decoding analyses identified in the localizer task. Decoding within an ROI vs. across the whole brain offers complementary advantages (in terms of sensitivity and generalizability across tasks) and we will utilize both strategies, considering positive results in either approach to be indicative of decodability.

According to GNW, decoding of category and location should be found in PFC for all seen trials and should generalize across tasks (AT-go-seen, AT-nogo-seen, dAT-seen), specifically in the 300–500 ms time-window. According to IIT, decoding of category and location should be maximal in posterior cortical areas for all types of seen trials, and decoding should generalize across tasks (AT-go-seen, AT-nogo-seen, dAT-seen) with a maximum in posterior cortical areas.

**3. Temporal dynamics of the neural differences between stimuli reported as seen vs. unseen.** We will track the neural dynamics of the difference between seen and unseen stimuli by running an ANOVA analysis with Visibility (seen/unseen) and Time (Early: 0–250 ms, Late: 250–500 ms). This analysis will not involve fMRI data due to the temporal precision required. We will test for a main effect of visibility and an interaction, with the latter testing a key prediction of GNW, i.e., a neural ignition for consciously perceived stimuli, reflected by a late (>250 ms) amplitude difference between dAT-seen and dAT-unseen trials. Prior to 250 ms, activation for dAT-unseen stimuli should be similar (if not equal) to dAT-seen activity, reflecting a fast feedforward sweep propagating from posterior to anterior cortices as well as local recurrent processing. The same is true for decoding of content. Therefore, late differences in amplitude in prefrontal areas between dAT-seen and dAT-unseen trials are expected, with little or no differences earlier in time. IIT does not make explicit predictions about the temporal dynamics differentiating seen from unseen conditions.

**4. Baseline differences between stimuli reported as seen vs. unseen.** Pre-stimulus baseline activity will be compared between dAT-seen and dAT-unseen trials. We will run a logistic regression with the categorical variables seen/unseen and the continuous variable amplitude to test GNW predictions, and the same logistic regression with the continuous variable's amplitude and synchrony between category selective and early visual areas to test IIT predictions.

GNW predicts higher pre-stimulus baseline activity in PFC for dAT-unseen than for dAT-seen trials. IIT predicts higher pre-stimulus baseline activity in category specific areas, and/or higher pre-stimulus baseline synchrony between category specific areas and lower visual cortices, in dAT-seen than in dAT-unseen trials.

*Pilot data*. This study was piloted for behavior and eye movement patterns. Results can be found on the OSF page (https://osf.io/6vgy3/). Those tests indicated an adequate number and approximately balanced (though variable between subjects) ratio of seen and unseen trials in the dAT condition ("dAT-seen" and "dAT-unseen"), and that nearly all stimuli are seen in the AT condition ("AT-seen"). The pilot eye tracking results demonstrated that subjects are able to maintain fixation while using peripheral attention to play the game; and that neither eye movements nor blink dynamics vary systematically between seen versus unseen trials. Finally, we conducted an analysis of low-level features of the video game and were able to rule out many potentially confounding variables related to the game dynamics and visual display, showing that by and large they did not contribute to the visibility of the stimulus.

## Data collection procedures for Experiments 1 and 2

**1. Functional Magnetic Resonance Imaging (fMRI).** Imaging will be conducted at the Yale Magnetic Resonance Research Center (MRRC) in New Haven and at the Donders Centre

for Cognitive Neuroimaging (DCCN), of Radboud University Nijmegen in Netherlands. Both centers have a Siemens 3T Prisma research scanner (Siemens, Erlangen, Germany) with high performance gradients (max. gradient strength 80mT/m, 200mT/m/s rise time, 100% duty cycle), 32-channel parallel imaging and a 32-channel head coil.

The systems are equipped with behavioral testing apparatus including visual display on a screen projected to a mirror fixed to the head coil, a button response system, and software for display and recording of behavioral tasks synchronized with the MRI data acquisition. At Donders Institute, stimuli will be presented on an MRI compatible Cambridge Research Systems BOLDscreen 32" IPS LCD monitor (resolution 1920 x1080 at 60Hz; viewing distance ~134cm). Psychology Software Tools Hyperion projection system (1920 x 1080 at 60Hz; viewing distance ~113cm) will be employed at Yale MRRC to project stimuli on the mirror fixed to the head coil.

Eye position will be monitored with an MR-compatible EyeLink 1000 Plus (SR Research Ltd., Ottawa, Canada) eye tracker. Only one eye will be recorded in the scanner: the left eye in Donders Institute and the right one in Yale. Pupil and corneal reflection will be sampled at 1000 Hz and analyzed to ensure that participants fixate at the accurate position. The eye tracker will be calibrated at the beginning of each session and repeated between runs if necessary.

Anatomical and functional images will be acquired on a 3T Prisma scanner, using a 32-channel head coil. Anatomical images will be acquired using a T1-weighted magnetization prepared rapid gradient echo sequence (MP-RAGE; GRAPPA acceleration factor = 2, TR/TE = 2300/3.03 ms, voxel size 1 mm isotropic, 8° flip angle). Functional images will be acquired using a whole-brain T2*-weighted multiband-4 sequence (time repetition [TR] / time echo [TE] = 1500/39.6 ms, 68 slices, voxel size 2 mm isotropic, 75° flip angle, A/P phase encoding direction, FOV = 210 mm, BW = 2090 Hz/Px). In addition, a single band reference image will be acquired before each run. To allow for signal stabilization the first three volumes of each run will be discarded. To correct for susceptibility distortions, additional scans using the same T2*-weighted sequence, but with inverted phase encoding direction (inverted RO/PE polarity) will be collected while the participant is taking rest at multiple points throughout the experiments.

**2. Magneto-Electroencephalography (M-EEG).** *Electrophysiological data acquisition.* M-EEG recordings will be acquired at the Centre for Human Brain Health (CHBH) of University of Birmingham in the United Kingdom, and at the Center for MRI Research of Peking University (PKU) in China. Both centers have a 306- channel, whole-head TRIUX MEG system from MEGIN (York Instruments; formerly Elekta). The MEG system comprises 204 planar gradiometers and 102 magnetometers in a helmet-shaped array. Simultaneous EEG will be recorded using an integrated EEG system and a 64-channel electrode cap. The MEG system is equipped with a zero boil-off Helium recycling system and the noise-resilient ARMOR sensors and placed in a shielded room (2 layers of mu-metal and 1 layer of aluminum). In order to covers the brain more homogeneously, the MEG gantry will be positioned at 68 degrees.

Prior to each experiment, MEG signals from empty room will be recorded for 3-minutes. We will also record 5-minutes of resting-state data for each participant. M-EEG signals will be sampled at a rate of 1 kHz and band-pass filtered between 0.01 and 330 Hz prior to sampling. The location of the fiducials and the positions of the 64 EEG electrodes will be recorded using a 3-D digitizer system (Polhemus Isotrack). A set of bipolar electrodes will be placed on the subject's chest (upper left and upper right chest position) to record the cardiac signal (ECG). Two sets of bipolar electrodes will be placed around the eyes (two located at the outer canthi of the right and left eyes and two above and below the center of the right eye) to record eye movements and blinks (EOG). Ground and reference electrodes will be placed on the back of the

neck and on the right cheek, respectively. The participant's head position inside the MEG system will be measured at the beginning and at the end of each run using four head position indicator (HPI) coils placed on the EEG cap. Specifically, the HPI coils will be placed next to the left and right mastoids and on left and right frontal areas. Their location relative to anatomical landmarks will be digitized with a Polhemus Isotrak System. During the measurement, high frequency (>200 Hz) signals are produced by those coils and the localization of these signals is used to estimate the head position in the sensor space. However, the interaction between the signals generated by these coils can be non-linear and produce some artifacts that are difficult to filter out. To avoid this issue, head position measurement will be performed only during resting periods (as opposed to continuously).

Anatomical *MRI data acquisition*. For each subject, a high resolution T1-weighted MRI data (3T Siemens MRI Prisma scanners (32 channel coil) with a resolution of 1 x 1 x 1 mm, 208 sagittal slices; field of view (FOV): 256 × 256 matrix) will be acquired before or after the MEG acquisition to later perform accurate source localization using individual-subject realistic head model.

*Behavioral setup*. In both centers, visual stimuli will be presented on a screen placed in front of the participant with a PROPixx DLP LED projector (VPixx Technologies Inc.) at a resolution of 1920 x 1080 pixels and a refresh rate of 120 Hz. The distance between the subject's eyes and the screen will vary across the labs (CHBH: 119 cm, PKU: 85) in order to archive in both setups a field of view of 36.6 x 21.2 degrees. In Experiment 2, sounds will be delivered through a set of MEG-compatible earphones (provided by MEGIN) connected to the audio interface of the MEG system. Participants will respond with both hands using two 5-button response boxes (CHBH: NAtA, PKU: SINORAD).

*Eye tracker*. In both centers, eye movements will be monitored and recorded from both eyes (binocular eye-tracking) using MEG compatible EyeLink 1000 Plus eye-tracker (SR Research Ltd., Ottawa, Canada). Nine-point calibration will be performed at the beginning of the experiment, and recalibrated if necessary at the beginning of each block. Pupil size and corneal reflection data will be collected at a sampling rate of 1KHz.

*Source localization*. Source localization will be performed using beamforming (LCMV or DICS) or minimum-norm estimation approach (MNE, dSPM). Source-level analyses will be performed in each subject, co-registered to individual anatomical MRIs using the digitized head surface and anatomical landmarks. Time-resolved multivariate decoding will be performed in the sensor space, from which the decoding weights are obtained for each sensor. Source localization will then be performed on the weight maps to project the decoding results to source space [61]. To verify the cortical source localization for the decoding results in M-EEG, we will examine whether face/object decoding is localized to the corresponding selective areas, e.g., FFA, PPA. In addition, we will consider using fMRI's decoding results to localize ROIs for M-EEG decoding analysis.

*Inter-subject alignment*. Individual data will first be aligned to a standardized brain template (i.e., the 'fsaverage' template) before the group analysis. We plan to use two inter-subject alignment methods and test which one is better during phase 2 (analysis development). One approach is to generate the forward (the leadfield) model with the spacing between the sources constrained to the MNI "fsaverage" brain (i.e., the same number of sources per area independently of the individual differences). The second approach is to create the leadfield model from the individual brain using an even spacing (i.e., the number of sources will depend on the size of each specific areas). Both approaches will be explored and compared during the analysis development stage.

**3. Invasive Electroencephalograghy (iEEG).** iEEG recordings will be obtained from patients with pharmacologically resistant epilepsy undergoing invasive electrophysiological

monitoring at the Comprehensive Epilepsy Center at New York University Langone Health Center, Brigham and Women's Hospital, Children's Hospital Boston, and Johns Hopkins Medical School and University of Wisconsin School of Medicine and Public Health.

*Electrophysiological data acquisition.* Brain activity will be recorded from intracranially implanted subdural platinum-iridium electrodes embedded in SILASTIC sheets (2.3 mm diameter contacts, Ad-Tech Medical Instrument and PMT Corporation). The decision to implant, electrode targeting, and the duration of invasive monitoring will be solely determined on clinical grounds and without reference to this or any other study. Macroelectrodes will be arranged as grid arrays (8 × 8 contacts, 10 or 5 mm center-to-center spacing), linear strips (1 × 8/12 contacts), or depth electrodes (1 × 8/12 contacts), or a combination thereof. Subdural electrodes will cover extensive portions of lateral and medial frontal, parietal, occipital, and temporal cortex of the left and/or right hemisphere. Recordings from grid, strip and depth electrode arrays will be made using a Natus Quantum (Pleasonton, CA), Xltek (San Carlos, CA) or a Blackrock system (Salt Lake City, UT) amplifier. Recordings are obtained continuously during the patients' stay in the hospital. All data will be stored with stimulus and timing markers permitting offline synchronization.

*Surface reconstruction and electrode localization.* Pre-surgical T1-weighted MRIs (no electrodes) and post-surgical CT scan (with electrodes) will be acquired for each patient and used to determine electrode locations, following the general methodology of Yang et al. (2012) or Dale et al. (1999) [62, 63] using the Freesurfer package (HU). For NYU, skull-stripped post-surgical CT images will be linearly co-registered to the pre-surgical MRI. Electrode locations will be extracted using the co-registered CT images, and projected to the reconstructed brain surface, generally following the procedure described by Yang and colleagues. MRI images will be nonlinearly registered to an MNI-152 template. The same transformation will be applied to the co-registered CT image in order to map the extracted electrode coordinates in Montreal Neurological Institute (MNI) space. A three-dimensional reconstruction of each patient's brain will be computed using FreeSurfer (http://surfer.nmr.mgh.harvard.edu).

At Harvard, the pial surface of each subject will be reconstructed using Freesurfer from the presurgical MRI. The Freesurfer function "recon-all" will be used, following the fully-automated directive ("-autorecon-all" flag). The CT images of the implanted electrodes will be localized to the MRI reconstruction using the iELVis package (Groppe et al., 2017) [64]. CT gantry tilt will be corrected using the dcm2niix package from www.nitrc.org. Electrode grid and strip orientation will be identified in the CT scan based on pre-surgical sketches and platinum marker guides (Ad-Tech, Racine, WI, USA). The electrode locations will then be summarized by mapping them onto one of 36 brain areas based on the parcellation of the Desikan-Killiany atlas (Desikan et al., 2006) [65] using the "recon-all" function in Freesurfer.

*Behavioral setup.* The experiment will be controlled and the stimuli will be presented on a Dell Precision 5540 laptop, with 15.6" Ultrasharp screen (screen size 357.27 x 235.47 mm2; resolution 1920x1080) running on Windows 10. The laptop will be positioned at 55–65 cm from the patient. Audio will be presented through loudspeakers.

*Eye tracker.* Eye-tracking data will be collected throughout the duration of the experiment using a Tobii-4C eye-tracker (New York University) and EyeLink 1000 Plus Camera (Harvard University and University of Wisconsin). Thirteen-point calibration will be performed several times during the experiment. The calibration will be performed at the beginning of the experiment, and recalibrated if necessary to meet precision requirements at the beginning of each mini-block. Pupil size and corneal reflection data will be collected at a sampling rate of 90Hz at New York University and at a sampling rate of 500Hz at Harvard university and University of Wisconsin. Only one eye will be recorded as determined by ocular dominance. The experiment will not be influenced by the Eye-tracking recording.

## Data management and sharing plan

Our plan to share neuroimaging, electrophysiological, behavioral and eye tracking data, and our management of intellectual property will be in accordance with the policies and guidelines of our institutions and NIH. All investigators involved in this project will adhere to NIH's Data Sharing Policy and Implementation Guidance of March 5, 2003 and NIH Grants Policy on Sharing of Unique Research Resources including the "Sharing of Biomedical Research Resources: Principles and Guidelines for Recipients of NIH Grants and Contracts" issued in December, 1999.

The investigators acknowledge their willingness to share data and materials stemming from this project in order to maximize impact and to accelerate discoveries to understand consciousness and its neural substrates. As soon as data have been acquired, data will be uploaded to the central cloud server of the project, hereby making it immediately available to all members of the team. Throughout the course of the 3 years that the project is expected to last, data will be shared with local colleagues at seminars and talks, and with the scientific community at large by posters and presentations at local, regional, national and international scientific meetings. Finally, data will be presented by publication to the widest audience possible.

Following publication of the primary adversarial collaboration manuscript, the research data will be made available to the scientific community through public repositories.

Specifically, as part of our resource sharing plan, we will develop a website on which the following resources pertaining to this proposal will be made publicly available:

1. Any Matlab (binaries)/ Python code used to analyze or pre-process the data

2. Any MRI pulse sequences (binaries) used to acquire the data

3. All (de-identified and PHI removed) MRI, neurophysiological, eye tracking and behavioral data adhering to the FAIR principle. Specifically, raw data will be shared both in native format and also in Brain Imaging Data Structure BIDS format (http://bids.neuroimaging.io). All metadata will be made available.

4. Any peer-reviewed article (depending on journal restrictions)

5. Any experimental code

In order to access or download files containing behavioral, eye tracking, neurophysiological or MRI data, users will have to register. As they register, they will agree to restrictions against attempting to identify study participants, restrictions on redistribution of the data to third parties, and to properly acknowledge the data resource. The neurophysiological recordings (clinical iEEG and experimental electrodes), relevant task data, electrode coordinates in MNI space and essential, de-identified clinical data using NINDS Common Data Elements (age, sex, duration of epilepsy, epilepsy etiology, preoperative imaging findings) and schematics of seizure onset areas will also be made available. Data will be converted to shareable data formats (BIDS). Machine-readable annotation of the task and Matlab/Python scripts to synchronize data streams will be included. All data will be reviewed prior to upload to ensure they contain no PHI. Data will be stripped of voice recordings and be HIPAA compliant. Data will be made available to the research community by sharing them through different steps in the analysis pipeline. We will share un-processed (DICOM), minimally preprocessed (NIFTI format) and final processed (NIFTI format) data. All custom code and analysis pipeline will also be shared.

## Discussion

We have embarked upon an ambitious and large-scale endeavor involving a dozen laboratories in three continents using an open science and adversarial collaborative process. The field has

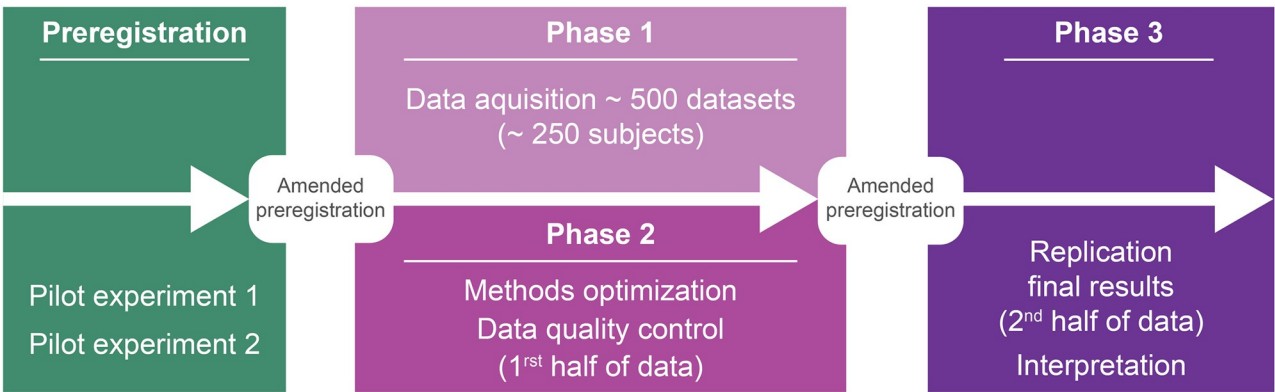

**Fig 4. Timeline for the adversarial collaboration protocol.** Number of subjects refers to the sum of subjects collected in the two experiments. Due to the COVID-19 pandemic of 2020–2021, phase 1 and 2 will be carried out in parallel and adjustments to the timing of some of the phases may be necessary.

already made substantial progress with a myriad of individual studies [e.g., 18, 66], but systematic tests of the theories' predictions in a large-scale, multimodal design have not yet been attempted. The present study protocol results from the first stage of this project–finding agreement among scientists with conflicting views for two meticulously planned studies, the associated instruments and analysis tools, as well as funding for such a costly enterprise. The two studies were extensively piloted (on more than 250 subjects; see OSF page). Those pilots established the adequacy of the experiments for yielding the desired behavioral and eye movement patterns, and served to rule out potential confounds. We are now in the process of collecting 500 datasets (250 subjects x 2 studies) using our chosen techniques—fMRI, M-EEG and iEEG—in phase 1 (Fig 4). This will complete the data acquisition phase. The first half of this massive data volume, that will reside in a secure server during the completion of the project, will be used to test the specific and detailed hypotheses described above, in phase 2. All aspects of the processing pipeline will be finalized and frozen before proceeding to the analysis of the second half of the data in phase 3, to assure the generalizability of the final results. These will then be published in platform papers, with all data, meta-data, stimulation protocols, and analysis code made freely available to anyone for further analysis or extension experiments.

GNW and IIT make several different fundamental assumptions about consciousness that are, at this point, challenging to directly test at the relevant level of granularity. In the two experiments outlined here, we focus on specific predictions that differentiate between these two theories regarding various aspects of the NCC that can be empirically addressed using contemporary brain recording techniques. Given the vast and ill-understood complexity of the brain, extant instrumental and biological variability across subjects and trials, and the distinct acquisition methods used, it is possible that no unambiguous answer may emerge from these experiments. At the least, we expect our results to challenge specific aspects of either or both theories and to further our understanding of conscious visual perception by tracking its footprints in the human brain.

## Acknowledgments

We are indebted to Heather Berlin, William Jaworski, Hakwan Lau and Cyriel Pennartz for insightful discussions during the two-day meeting organized by the Templeton World

Charity Foundation at the Allen Institute, Seattle in March 2018; to Hakwan Lau for helping conceptualize the proposed studies; to Johannes J. Fahrenfort, Sebastien Marti and Cyriel Pennartz for insightful comments on earlier versions of this manuscript and to Dawid Potgieter for spearheading this initiative and coordinating the process leading up to the submission of this preregistration. We further thank Shai Fischer, Rony Hirschhorn, Dooyoing Kim, Andrew Kyroudis, Alex Lepauvre and Kevin Ortego for running the pilot experiments and Felix Bernoully for developing the stimuli (Letters and Symbols) used in Experiment 1. Experiment 1 was programmed by Yoav Roll, Alex Lepauvre, Aya Khalaf, and Katarina Bentz. The video game used in Experiment 2 was developed by Konstantinos Vasileiadis and Aris Semertzidis with assistance from Nikos Gregos. Special thanks to our expert advisors Sylvain Baillet, Radoslaw Cichy, Michael Kahana, and Essa Yacoub, to Fosca Ai Roumi for advice on GNW theory, and our Scientific Coordinator Tanya Brown for coordinating critical aspects of the Cogitate consortium.

## Author Contributions

**Conceptualization:** Lucia Melloni, Liad Mudrik, Michael Pitts, Hal Blumenfeld, Melanie Boly, David J. Chalmers, Francis Fallon, Theofanis I. Panagiotaropoulos, Stanislas Dehaene, Christof Koch, Giulio Tononi.

**Data curation:** Lucia Melloni, Liad Mudrik, Michael Pitts, Oscar Ferrante, Urszula Gorska, Rony Hirschhorn, Aya Khalaf, Csaba Kozma, Alex Lepauvre, Ling Liu, David Mazumder, David Richter, Hao Zhou.

**Formal analysis:** Katarina Bendtz, Oscar Ferrante, Urszula Gorska, Rony Hirschhorn, Aya Khalaf, Csaba Kozma, Alex Lepauvre, Ling Liu, David Richter.

**Funding acquisition:** Lucia Melloni, Liad Mudrik, Michael Pitts.

**Investigation:** Lucia Melloni, Liad Mudrik, Michael Pitts, Katarina Bendtz, Oscar Ferrante, Urszula Gorska, Rony Hirschhorn, Aya Khalaf, Csaba Kozma, Alex Lepauvre, Ling Liu, David Mazumder, David Richter, Hal Blumenfeld, Sasha Devore, Floris P. de Lange, Ole Jensen, Gabriel Kreiman, Huan Luo.

**Methodology:** Lucia Melloni, Liad Mudrik, Michael Pitts, Katarina Bendtz, Oscar Ferrante, Urszula Gorska, Rony Hirschhorn, Aya Khalaf, Csaba Kozma, Alex Lepauvre, Ling Liu, David Richter, Hal Blumenfeld, Melanie Boly, David J. Chalmers, Sasha Devore, Francis Fallon, Floris P. de Lange, Ole Jensen, Gabriel Kreiman, Huan Luo, Theofanis I. Panagiotaropoulos, Stanislas Dehaene, Christof Koch, Giulio Tononi.

**Project administration:** Lucia Melloni, Liad Mudrik, Michael Pitts, Hal Blumenfeld, Sasha Devore, Floris P. de Lange, Ole Jensen, Gabriel Kreiman, Huan Luo.

**Resources:** Lucia Melloni, Liad Mudrik, Michael Pitts, Katarina Bendtz, Urszula Gorska, Rony Hirschhorn, Alex Lepauvre, Hal Blumenfeld, Sasha Devore, Floris P. de Lange, Ole Jensen, Gabriel Kreiman, Huan Luo.

**Software:** Katarina Bendtz, Oscar Ferrante, Urszula Gorska, Rony Hirschhorn, Aya Khalaf, Csaba Kozma, Alex Lepauvre, Ling Liu, David Richter.

**Supervision:** Lucia Melloni, Liad Mudrik, Michael Pitts, Hal Blumenfeld, Sasha Devore, Floris P. de Lange, Ole Jensen, Gabriel Kreiman, Huan Luo, Giulio Tononi.

**Validation:** Lucia Melloni, Liad Mudrik, Michael Pitts, Katarina Bendtz, Oscar Ferrante, Urszula Gorska, Rony Hirschhorn, Aya Khalaf, Csaba Kozma, Alex Lepauvre, Ling Liu, David

Mazumder, David Richter, Hal Blumenfeld, Sasha Devore, Floris P. de Lange, Ole Jensen, Gabriel Kreiman, Huan Luo.

**Visualization:** Lucia Melloni, Liad Mudrik, Michael Pitts, Katarina Bendtz, Rony Hirschhorn, Alex Lepauvre.

**Writing – original draft:** Lucia Melloni, Liad Mudrik, Michael Pitts, Ling Liu, David Richter, Christof Koch.

**Writing – review & editing:** Lucia Melloni, Liad Mudrik, Michael Pitts, Katarina Bendtz, Oscar Ferrante, Urszula Gorska, Rony Hirschhorn, Aya Khalaf, Csaba Kozma, Alex Lepauvre, Ling Liu, David Mazumder, David Richter, Hao Zhou, Hal Blumenfeld, Melanie Boly, David J. Chalmers, Sasha Devore, Francis Fallon, Floris P. de Lange, Ole Jensen, Gabriel Kreiman, Huan Luo, Theofanis I. Panagiotaropoulos, Stanislas Dehaene, Christof Koch, Giulio Tononi.

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
