## [Decision Letter · Decision Letter 0]

17 Jan 2022

PONE-D-21-36359An adversarial collaboration protocol for testing contrasting predictions of global neuronal workspace and integrated information theoryPLOS ONE

Dear Dr. Pitts,

Thank you for submitting your manuscript to PLOS ONE. After careful consideration, we feel that it has merit but does not fully meet PLOS ONE’s publication criteria as it currently stands. Therefore, we invite you to submit a revised version of the manuscript that addresses the points raised during the review process.

Both the reviewers and I feel that this project is of high value. We are asking for only a minor revision of your manuscript, in keeping with the requirement of PLOS ONE for methodological rigour and completeness. Reviewer 2 has a number of suggestions about how to make the project even better, and I would like to see many, if not most, of these incorporated into your revision (and perhaps even into your preregistered analysis plan). I believe phase 2 of your analysis allows you to make revisions in your process before undertaking phase 3, so this should be possible. I particularly hope that you will seriously consider doing the Bayesian analysis suggested both by reviewer 2 and by me in my additional comments. 

We look forward to receiving your revised manuscript.

Kind regards,

Lawrence M Ward

Academic Editor

PLOS ONE

Journal Requirements:

“This work could not have been possible without the continuous and generous support of the Templeton World Charity Foundation at the different stages of this project to Christof Koch, Michael Pitts, Liad Mudrik and Lucia Melloni, as well as of the Max Planck Society (Lucia Melloni). This project is made possible through the support of a grant from Templeton World Charity Foundation, Inc (TWCF0389) to Lucia Melloni and (TWCF0378) to Liad Mudrik and (TWCF0345) to Michael Pitts. The opinions expressed in this publication are those of the author(s) and do not necessarily reflect the views of Templeton World Charity Foundation, Inc.”

“The authors have declared that no competing interests exist.”

Additional Editor Comments:

This paper should stand on its own without the need to refer to the preregistered methods and analyses, at least so far as a first reading for the overall sense of the project and the sufficiency of the predictions and data analyses are concerned. My extra (to the reviewers) comments mostly relate to this requirement. Moreover, you should write a sentence or two earlier on stating that readers who wish to view specific analysis plans should see the prereg methods and giving the link there as well as when you do that later in the paper.

Re the theories: please give a more extended description of the theories - at least another paragraph or two for each, in which critical terms are defined and a few more references are cited to support choosing these two theories to test. For example, the anatomy of the long-range projections critical to GWS could be described, and a definition of the “cause-effect structure” of IIT could be given (both mentioned by reviewer 2).

This project is ideal for a Bayesian approach (see reviewer 2 comments) in which the various experiments can provide evidence for or against (or neutral) for each of the theories. At the end we should have a posterior probability that each of the theories is a good one for the NCC. This won’t be a final answer, of course, at least partially because other theories predict some of the same outcomes. But at least there should be a sense of which of the two tested here, if either, is more promising. Mixed results, of course, might favour neither theory, and this too would be a useful outcome, with final odds near 1:1. I strongly encourage you to at propose (and perform) a complementary analysis of the results based on an iterative Bayesian approach. You can begin with a neutral prior and see where the data take you.

Re multivariate decoding in Experiment 1: please mention here which decoding scheme will be used, especially for M-EEG (linear support vector machine, as mentioned in preregistered methods- could consider using a Bayesian SVM?). Also, please describe how cortical sources will be localized (i.e., if and how you will move from sensor space to source space, via independent component analysis, beamforming, …)? Also, how will inter-subject alignment be accomplished for group analyses?  Please state that here and provide a citation. Moreover, decoding M-EEG, especially re cortical sources, is not nearly as advanced as that of fMRI or of iEEG (e.g., work of Knight lab). Results of machine learning even for sensor space can be controversial (i.e., cf. Cruse et al, 2011, The Lancet vs Goldfine, et al, 2013, The Lancet). Yet hypotheses re timing (300-500 msec window) depend on either iEEG or M-EEG. Please state how cortical source decoding of M-EEG data will be verified (via co-registered fMRI?).

Expt 1, analysis 3: You propose to determine interareal information sharing using the weighted phase lag index (general methods). Will this be between sources? How will these be determined from M-EEG data? Why not also use Granger causality or transfer entropy to obtain directed information flow?

Re MEG head position: MEG is very sensitive to head position and head position can vary considerably during runs. Measuring at the beginning and end of runs is the old way to control for head position, with reruns being necessary when head position varied by more than a small amount. Modern machines usually can measure head position much more often and record it so that compensation can be made computationally and all data retained. Why is this not being done here?

Reviewers' comments:

Reviewer's Responses to Questions

**Comments to the Author**

1. Does the manuscript provide a valid rationale for the proposed study, with clearly identified and justified research questions?

Reviewer #1: Yes

Reviewer #2: Yes

2. Is the protocol technically sound and planned in a manner that will lead to a meaningful outcome and allow testing the stated hypotheses?

Reviewer #1: Yes

Reviewer #2: Partly

3. Is the methodology feasible and described in sufficient detail to allow the work to be replicable?

Reviewer #1: Yes

Reviewer #2: Yes

4. Have the authors described where all data underlying the findings will be made available when the study is complete?

Reviewer #1: Yes

Reviewer #2: Yes

5. Is the manuscript presented in an intelligible fashion and written in standard English?

Reviewer #1: Yes

Reviewer #2: Yes

6. Review Comments to the Author

You may also provide optional suggestions and comments to authors that they might find helpful in planning their study.

Reviewer #1: This is interesting, stimulating and very important paper which aims to advance our knowledge on the neural correlates of consciousness (NCC). Specifically, it outlines a collaboration protocol for testing contrasting predictions of global neuronal workspace (GNW) and integrated information theory (IIT) in relation to phenomenon of consciousness.

The project will focus on conscious vision in human subjects and involve two experiments, using three complementary methods: functional magnetic resonance imaging (fMRI), simultaneous magnetoencephalography & electroencephalography (M-EEG) and intracranial electroencephalography (iEEG). All relevant hypotheses, predictions, and planned methods are already preregistered and published at OSF: https://osf.io/mbcfy/

To my opinion and considering the format of the paper – Protocol – the paper is ready for publication.

Reviewer #2: I enjoyed reading this comprehensive and accessible description of an adversarial collaboration to test GNW against IIT predictions of the neuronal contents of consciousness. This is an intriguing endeavour and there has clearly been an enormous amount of work behind-the-scenes, in putting this international and interdisciplinary project together. I thought that you did a great job in summarising the issues you have been contending with.

I am not sure whether it is appropriate to critique the conclusions of your discussions at this stage; however, I have some comments about your data analysis – which I will review from a practical perspective. Otherwise, I have a few minor points that might help clarify the presentation of your proposal for the general reader.

Major points

Your motivation, experimental design and protocols are all very impressive. However, this is not reflected in the quality of your proposed data analysis. Most of your analytic approaches may not be fit for purpose to fulfil the objective of the adversarial collaboration. The current data analysis streams can be critiqued at two levels:

First, in testing the predictions of two or more hypotheses, you will need to present the evidence for both hypotheses, in each dataset. The evidence is just the likelihood of the data, under GNW or IIT predictions. This means you need to present Bayes factors (as opposed to classical statistics). Bayes factors will then allow you to assess the evidence for each hypothesis and, crucially, be able to say when there is no evidence for one over the other, given the data at hand (if the Bayes factors are suitably small). This is a key point that contrasts with classical inference, where one can never accept the null hypothesis of no difference. You may want to discuss issue with people who are experienced in Bayesian analysis of fMRI and (i)EEG data – and identify a coherent analysis stream that allows you to compute the requisite marginal likelihood or model evidence, under your carefully identified hypotheses.

The second level at which your data analysis falls short is the use of decoding and RSA. Implicit in the use of decoding are constraints that could make your job very difficult. One obvious constraint is that decoding is generally limited to testing for main effects (i.e., targets versus non-targets, seen versus unseen, et cetera). This is a problem because you need to test for interactions: for example, the interaction between stimulus category and task – or between cortical area and task.

I imagine that you hope to use decoding accuracy (or correlations between vectorised RSA matrices) as summary statistics – and then test for interactions using classical inference (e.g., t-tests). The problem is that these kinds of summary statistics are notoriously inefficient (by the Neyman-Pearson lemma).

Furthermore, some of your hypothesis tests rest upon accepting the null. For example, "GNW predicts that conscious content will be decodable from the global workspace, independently of task”. However, using classical inference, you cannot accept the null hypothesis of no difference between classification accuracy between task-relevant and task-irrelevant conditions.

The only analysis that seemed appropriate was the ANOVA of responses to stimuli reported as seen and unseen and an interaction with time, “with the latter testing a key prediction of GNW”. If you could use this kind of analysis throughout, I think you would be on a much stronger footing.

I make these observations because, I can see – in a year or so's time – a beautifully articulated report that ends up with a succession of: "Unfortunately, we could not reject the null hypothesis…"

I do not know whether it is appropriate to address these issues in the current paper. Perhaps it would be sufficient to add some qualifications to the discussion along the above lines?

Minor points

On page 2, you say that neurons in prefrontal and cingulate cortices are connected through long-range excitatory axons to high-level sensory areas. While this is true, you attribute the sources of these connections to pyramidal cells of layer 2 and 3. This goes against the usual conception of top-down or descending excitatory connections to sensory levels, which are thought to be deep pyramidal cells. Perhaps you could nuance this sentence or provide some references (this issue is relevant because superficial and deep pyramidal cells contribute differentially to some of the signals you will be analysing).

In the description of integrated information theory, it may be useful to provide some heuristic descriptions of the technical terms – perhaps in a glossary? For example, what is a “cause-effect structure”. More specifically, what is meant by a “structure” in this setting: is it a correlation structure, an adjacency matrix, a conditional probability et cetera? Structures seem to be closely related to the notion of a "substrate". It is the substrate meant to mean neuronal activity or connectivity? Finally, what is meant by "neuronal mechanisms" are these neuronal dynamics, changes in synaptic efficacy or something more abstract? In short, a little primer on the constructs used by IIT would be useful.

On page 4, could you replace "Our study focuses on the prefrontal cortex (PFC) as part of the NCC" with "Our study focuses on the NCC in the prefrontal cortex (PFC)".

On page 4, you talk about "activity-silent” states under the GNW. Can I suggest you remove phrase “activity silent”: otherwise, you might confuse people; who will be asking how can you look for the neural correlates of consciousness when they are silent? You could say something like:

"GNW rests upon neurophysiologically plausible belief updating – of the kind associated with predictive coding – with a well-defined dynamics over the conscious experience (e.g., characteristic onset and offset responses). IIT does not share the same commitments and posits that the pattern of neural activity should persist over the duration of a conscious experience."

Please capitalise “covid” in COVID-19.

In describing your experimental designs, it would be helpful to use a more formal description, so that people can get their heads around the factors. For example,:

“Experiment 1 is a nested factorial design with three factors; namely, stimulus category (face, object, letter versus false font), Target category (face versus letter), and duration (500, 1000 one thousand 500 ms). We were particularly interested in interaction between stimulus and target category; i.e., task relevance and the simple main effect of target versus no target in task relevant responses …”

On page 10, you say that GNW predicts patterns of activity in the first time window to differ from patterns in later time windows (in contrast to IIT). You then say:

"In RSA, high similarity should be found between content specific patterns of activity in the first three time windows independent of any initial transient. This activity should differ from that found during the fourth time window when the stimulus is no longer presented."

I suspect many readers will find this difficult to unpack because you seem to be talking about [dis]similarities of similarities. It would be simpler and more efficient to simply test for interactions between content and time windows using standard multivariate procedures (i.e., canonical variates analysis – CVA). CVA is a general linear model that can be regarded as a multivariate generalisation of an ANOVA. You can then test directly for the different hypotheses about time specific responses in spatially distributed patterns. I mention this because open source CVA code is available (e.g., spm_CVA.m) that returns approximations to log evidence (based on the AIC and BIC).

I hope that these comments help should any revision be required.

7. PLOS authors have the option to publish the peer review history of their article (what does this mean?). If published, this will include your full peer review and any attached files.

Reviewer #1: No

Reviewer #2: **Yes: **Karl Friston

---

## [Author Response · Author response to Decision Letter 0]

5 Apr 2022

Responses to editor's and reviewers' comments are uploaded in a separate document

---

## [Editor Report · Decision Letter 1]

3 May 2022

An adversarial collaboration protocol for testing contrasting predictions of global neuronal workspace and integrated information theory

PONE-D-21-36359R1

Dear Dr. Pitts,

Thank you for your excellent revision. We’re pleased to inform you that your manuscript has been judged scientifically suitable for publication and will be formally accepted for publication once it meets all outstanding technical requirements.

Kind regards,

Lawrence M Ward

Academic Editor

PLOS ONE
---

## [Editor Report · Acceptance letter]

22 Sep 2022

PONE-D-21-36359R1 

An adversarial collaboration protocol for testing contrasting predictions of global neuronal workspace and integrated information theory 

Dear Dr. Pitts:

I'm pleased to inform you that your manuscript has been deemed suitable for publication in PLOS ONE. Congratulations! Your manuscript is now with our production department. 

Kind regards, 

on behalf of

Dr. Lawrence M Ward 

Academic Editor

PLOS ONE